# Patient involvement in the biopsychosocial integrated primary care model: A qualitative study in three health districts of South Kivu, Democratic Republic of Congo

Bertin Mutabesha Kasongo[1,2]*, Christian Eboma Ndjangulu Molima[1], Gérard Jacques Mparanyi[3], Samuel Lwamushi Makali[1,4], Pacifique Lyabayungu Mwene-Batu[1], Albert Mwembo Tambwe[2], Hermès Karemere[1,3], Ghislain Balaluka Bisimwa[1,5,6], Abdon Mukalay wa Mukalay[2,7]

1 Ecole Régionale de Santé Publique, Catholic University of Bukavu, Bukavu, Democratic Republic of Congo, 2 School of Public Health, University of Lubumbashi, Lubumbashi, Democratic Republic of Congo, 3 Faculty of Pharmaceutical Sciences and Public Health, Official University of Bukavu, Bukavu, Democratic Republic of Congo, 4 Centre de recherche sur les politiques et systèmes de santé (CR3-POLISSI), Ecole de santé publique, Université libre de Bruxelles, Brussels, Belgium, 5 Centre de Recherche en Sciences Naturelles, Lwiro, Democratic Republic of Congo, 6 Université du Cinquantenaire, Lwiro, Democratic Republic of Congo, 7 Unité d'Epidémiologie Clinique et Pathologies Tropicales, Faculté de médecine, Université de Lubumbashi, Lubumbashi, Democratic Republic of Congo

* bkas2504@gmail.com

## Abstract

According to the World Health Organization, involving patient in healthcare provision and decision-making about their health is a key factor in ensuring healthcare quality. This study explores patients' involvement in choosing their care strategies, their responsibility for holistic care, and their ability to support biopsychosocial model of care. This qualitative research was conducted in three health districts in South Kivu province, eastern Democratic Republic of Congo, which benefit from interventions on chronic disease management. From February to April 2024, 27 individual interviews were conducted with members of chronic disease clubs in 6 health areas using a tool inspired by the International Alliance of Patients' Organizations' Declaration of Patient-Centered Care. The content of the interviews was analyzed using an Inductive Content Analysis approach. Five categories emerged from the interviews regarding patient involvement in the biopsychosocial model of care. Patients' participation in care revealed collaborative rapport between them and providers, the partnership, and the patient Clubs' involvement in community activities (home visits, awareness-raising). Patients' empowerment and decision-making were observed through their responsibility for care, their choice of treatment options, and the role of Clubs in care planning. Patient support systems included their preventive and promotional activities, the adaptation of care to their needs, the entourage, the Clubs, and their psychologist. Patient education and lessons learned from caregivers fostered behavior change, objectified by the development of skills (cognitive, social, and emotional).

**Data availability statement:** The minimal dataset underlying the findings of this study—including the final codebook, thematic matrix, and anonymized illustrative excerpts—is publicly available at Zenodo: [https://doi.org/10.5281/zenodo.17557004]. Full interview transcripts cannot be shared publicly because of confidentiality and ethical restrictions approved by the Ethics Committee of the Catholic University of Bukavu (UCB/CIES/NC/002/2024). Qualified researchers may request additional access through the corresponding author (bkas2504@gmail.com) or via the Ethics Committee of the Catholic University of Bukavu (vumilia.nakabanda@ucbukavu.ac.cd).

**Funding:** The authors received no specific funding for this work.

**Competing interests:** The authors have declared that no competing interests exist.

Suggestions formulated are mainly concentrated on humanizing care and services, and psycho-financial support for patients. Healthcare systems should consider all five dimensions identified in this study when defining policies and integrating biopsychosocial care. Strengthening these elements across the individual, provider, and community levels can enhance holistic patient involvement, promote self-management, and improve the quality and accessibility of care for chronic disease management in resource-limited settings.

## Introduction

Involving patients in providing care and decision-making about their health is a key factor in improving service delivery and ensuring quality of healthcare [1,2]. Involving the person in their own care means allowing him or her to participate autonomously in defining the healthcare policies that concern him or her (considering his needs, preferences, and independence), as well as in implementing and monitoring health services [3–6]. This aligns with person-centered care (PCC), which emphasizes patient's needs, preferences, and empowerment, while providing the education necessary for decision-making [7]. One of the core components of this PCC is the biopsychosocial (BPS) model of care, proposed by George Engel in 1977, in which care is provided comprehensively (holistically), according to biological, psychological, social, and spiritual dimensions, while respecting individual needs and values [8–10]. From this BPS perspective, involving person in the care process is crucial to achieving positive health outcomes.

Various factors influencing patient involvement in care have been identified and explored worldwide. These include patients' health education and knowledge, their attitudes, beliefs, and values, their experiences of healthcare services, the provider-patient relationship (as partners in care), support groups [11–15].

In addition to the biopsychosocial perspective, patient involvement in care can also be interpreted through the Social Ecological Model (SEM), which highlights the multiple, interacting layers that shape health behavior and participation in care. According to this model, individual actions and decisions are influenced not only by personal attributes and immediate social networks (such as family and support groups) but also by broader community and societal contexts, including health system organization, cultural norms, and policy environments. Patient involvement and empowerment depend on factors operating across these interconnected levels (from the individual and interpersonal to the community and societal spheres) each contributing to the enabling or constraining conditions of the biopsychosocial model of care [16,17].

Some studies have shown the benefits of involving patients in decision-making about their healthcare. They have revealed that patients who are involved in decision-making in care express a sense of well-being and self-satisfaction [18]. Others have reported improved interaction and communication between providers and patients [19], patient adherence to treatment [20], rapid adoption of healthy behaviors, and better self-management of illness through active participation in self-monitoring

activities [21]. Involving patients and their social supports has also demonstrated its importance in improving the quality of life and in managing certain health conditions that require the BPS model of care, such as chronic pain management [22].

In the context of low-income countries, particularly in Sub-Saharan Africa, patient participation and involvement in decision-making about their care are crucial. These efforts help to improve both individual health and the performance of health systems, which remain fragile. In some situations, this involvement is limited to the design of services and health research [23], or constrained by patients' lack of awareness of their right to participate in decision-making about their care, especially among those with lower levels of education [24]. Other experiences have yielded positive results in the provision of person-centered care, such as in the treatment of HIV [25]. Some authors have also demonstrated patient satisfaction and adherence to treatment resulting from the support obtained through social groups to which they belong [26].

Some of these experiences provided strategies for involving patients in their care, thereby facilitating the achievement of positive outcomes, particularly in mental health [27], where the BPS model of care also requires consideration of spiritual aspects [28]. Involving patients in their care also requires empowerment, as shown by studies analyzing the barriers and facilitators to patient empowerment in controlling their disease [29,30].

In the Democratic Republic of Congo (DRC), particularly in its Eastern region, where the health system faces complex challenges and crises (including conflict, violence, and widespread poverty, that further affect physical, mental, and social health), involving patients in the BPS care they receive is crucial. Such involvement strengthens collaboration among providers, communities (social groups), and patients, offering them care adapted to their needs across biological, psychological, and socio-cultural dimensions, thereby improving healthcare quality. Various studies have analyzed patients' involvement in decision-making about their health and healthcare services in specific contexts [31,32], as well as their perceptions and satisfaction with the holistic care offered [33,34]. However, to our knowledge, patient involvement in the BPS model of care remains poorly documented in the DRC.

Therefore, this study explores patients' involvement in choosing their care strategies, their responsibility for holistic care, and their ability to support this BPS model.

## Methods

### Study settings

**The organizational framework of the health system in the DRC (operational level).** The study was conducted in South Kivu Province, DRC, specifically in three Health Districts (HD), as shown on the provincial health map (Fig 1).

The DRC's healthcare system is divided into three levels: central or national, intermediate or provincial, and peripheral or operational. Our study was conducted at the operational level, which is composed of HDs.

This level is responsible for implementing Primary Health Care (PHC) strategies. It comprises the HDs (each generally with a General Reference Hospital), which are subdivided into Health Areas (HAs), each with a Health Center (HC). The HDs operate under the coordination of the HD Management Team, which oversees the planning, implementation, monitoring and evaluation of all HD activities. These are operationalized at the HA level by the HC team, consisting of the HC Head Nurse and other care providers. The HC serves as the point of contact between the health system and the population through community participation structures, direct community representatives (Health Area Development Committee – HADC, which is one of the HC's co-management structures, Community Animation Cell – CAC; Community Health Workers – CHWs) and other structures are known as "Community-Based Organizations – CBO." CHWs are organized into CACs at the village level (multi-sectoral community participation structure at the peripheral level), and each CAC is represented at the HADC level by its president [35,36].

**Geographical and health context of the HDs.** The three HDs concerned by the study benefit from the support of "Louvain Coopération", a Belgian academic non-governmental organization (NGO) of the "Université Catholique de

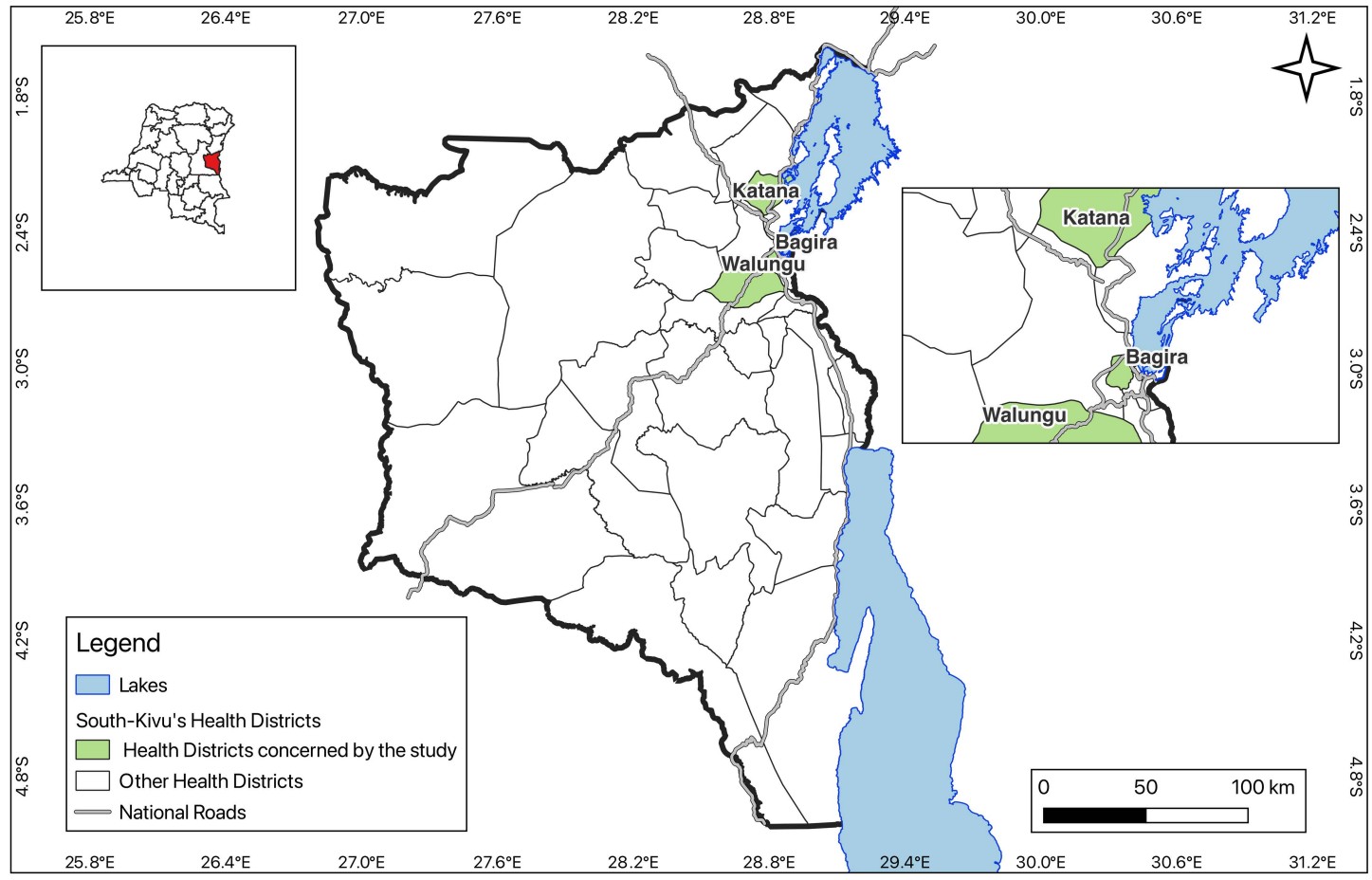

Source: HumDada.org; South-Kivu PHD; ERSP/UCB, B.Kasongo 2025

**Fig 1. The three study HD on the health map of South Kivu province.** Base map source: *Democratic Republic of Congo – Health district boundaries* (https://data.humdata.org/dataset/zones-de-sante-rdc) licensed under **ODbL**. Provincial Health Division of South Kivu (South-Kivu PHD), Figure created by the authors (B. Kasongo, ERSP/UCB, 2025).

Louvain" (UCLouvain), as part of the Non-Communicable Diseases Program (NCDP) from 2022 to 2026. This project focuses on managing diabetes, hypertension, and mental health as a gateway to BPS care.

The following table (Table 1) presents some information on the health of these three HDs.

This program covers four HAs in each of these three HDs. For this study, six of the twelve HAs benefiting from this support were selected based on reasoned choice and accessibility, with 2 HAs per HD (Lumu and Nyamuhinga HAs in the Bagira HD, Birava and Kabushwa HAs in the Katana HD, and Bideka and Lurhala HAs in the Walungu HD).

**Table 1. Description of health districts concerned by the study.**

| Health District | Population (inhabitants) | Health Center Number | Hospital | Rates of Utilization of Curative Services (%)* |
|---|---|---|---|---|
| Bagira (urban) | 162,186 | 8 | 3 | 32.8 |
| Katana (rural) | 267,504 | 20 | 4 | 72.1 |
| Walungu (rural) | 318,610 | 23 | 3 | 32.5 |

***(New Cases/Total Population) x 100. The health standard for the utilization rate of curative services is ≥ 50%.** (Source: health pyramid of South Kivu and District Health Information Software – DHIS-2, 2023)* [37]

**Conceptual framework for patient involvement in the BPS model of care.** This study was based on the conceptual framework of patient involvement in the BPS model of care, inspired and adapted from Carman KL et al. and Harrison SR & Jordan AM [38,39]. This framework is represented in the following figure (Fig 2).

From a BPS perspective in direct care, patients' involvement in care and decision-making about health depends on three interacting and mutually influencing components: the individual, the healthcare provider, and the community (social groups).

- Individual: The person is responsible for their own health. In this context, "empowerment" (the process of acquiring power, which helps the individual or community to take control of their own lives, with the prospect of change and improvement) is a key element of the BPS model [40–42]. The person is regarded as an expert on their own situation and body, which is essential for improving the quality of care. Successful treatment and recovery depend on participation, involvement, and acceptance in the provision of care [2,3]. Elements directly related to the individual also include biological and genetic characteristics, personal and social skills, lifestyle habits and behaviors, and socioeconomic factors [43].

- The healthcare provider: the interaction between patient and provider is crucial in this process of involvement in care decision-making. Empathy and compassion must characterize providers in this partnership of care [2,38]. Beyond clinical competence, providers' communication should integrate deeper relational attributes such as shared compassion, emotional arousal, aesthetic pleasure, simplification, and asymmetry, which reinforce trust and mutual understanding.

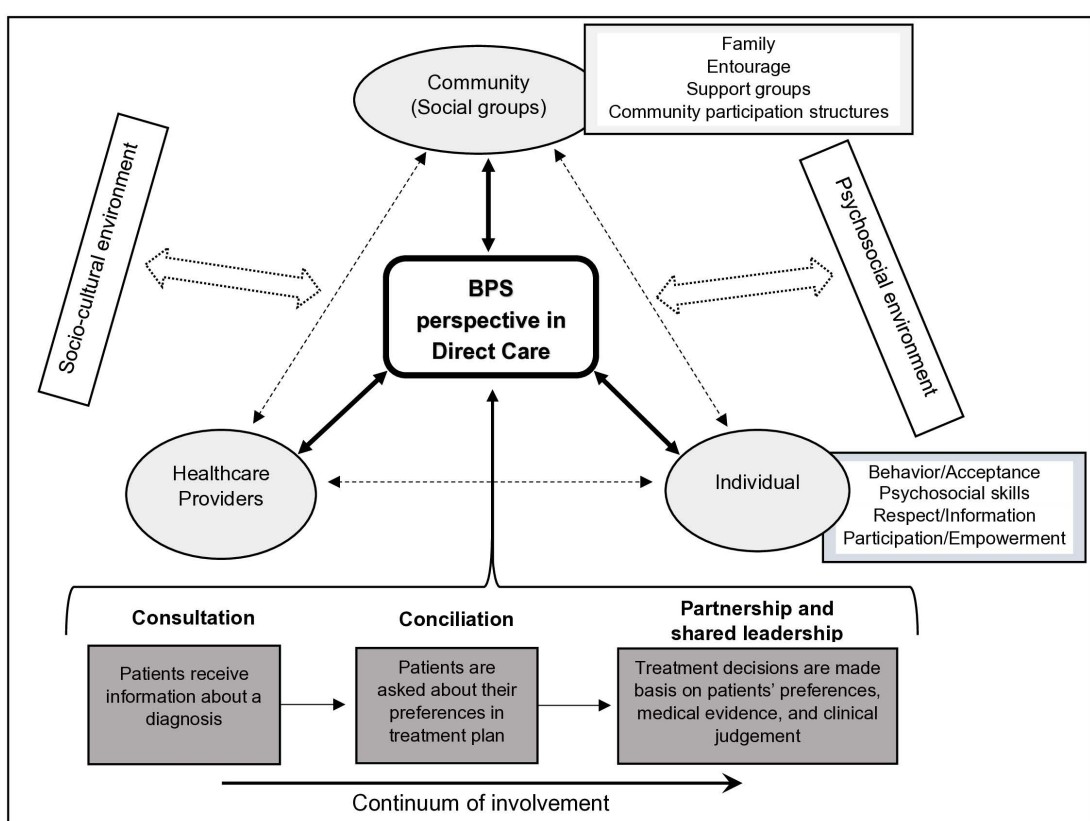

**Fig 2. Conceptual Framework for Patient Involvement in the BPS Model, inspired and adapted from Carman KL et al. and Harrison SR & Jordan AM [38,39].**

These elements align with the Sadharaṇikaraṇ Model of Communication, which emphasizes sahridayata (a shared sense of empathy and emotional connection between communicator and receiver). In health contexts, this model suggests that providers and patients co-create meaning through affective resonance, even within asymmetric relationships, by simplifying messages and fostering emotional harmony [44,45]. At the front line of care, in the management of chronic diseases, multidisciplinary should also be encouraged [46,47].

- Community (Social groups): Here, we have all the community groups and social networks (family, entourage, various associations or patient groups such as diabetic and hypertensive clubs), as well as community participation structures such as Health Area Development Committee, Community Animation Cell, Community-Based Organization, all of which act in immediate support of the individual, in collaboration with healthcare providers. The patient lives with other members of these social groups, in a sociocultural and psychosocial environment that constantly influences them. These elements need to be considered in care, especially in psychosocial support, a key component of the BPS model of care [48,49].

The form of the individual's involvement in decision-making and care according to the BPS model depends on the type of individual-care provider and individual-community relationship (from consultation to partnership and shared leadership), as shown in the figure (Fig 2).

### Study design

This qualitative study used a phenomenological approach within an interpretative paradigm [50,51]. The latter was chosen to convey the subjective experiences of the interviewees, employing in-depth interviews and content analysis to capture the richness and diversity of their experiences. Using this approach allowed us to examine and interpret how people with chronic diseases (particular people) understand their involvement in care (specific phenomenon), based on their experiences with the implementation of the BPS model (given context) [51,52].

### Study participant selection

Participants were selected through convenience sampling. A sample of 5 participants per HA was predefined for reasons of convenience and feasibility. As mentioned above, these HAs benefit from support in the management of chronic diseases under the NCDP. In these areas, some patients have organized themselves into clubs (diabetic and hypertensive clubs) where various activities are conducted for their care, including health education, awareness-raising, experience sharing, and mutual assistance. We included patients who were members of these clubs (which are considered patient support groups) provided they had been members for at least one-year prior the inquiry and actively participated in club activities. This approach was intended to help us better understand the dynamics surrounding the BPS model and the interaction between the three components of our conceptual framework: the individual, the healthcare providers/services, and the social groups (specifically these support groups). Healthcare providers assisted in identifying participants who met the inclusion criteria, and the investigators subsequently contacted them to participate in the study. A total of 27 patients accepted and consented to participate in the study. It should be noted that this sample made it possible to obtain a saturation of the information collected. Data saturation was assessed by the principal investigator (BMK) in consultation with the supervising researchers, once no new codes emerged during the last three interviews. The decision was made collectively after reviewing coded transcripts and confirming that subsequent interviews yielded repetitive information.

### Data collection

We conducted individual semi-structured interviews with the selected participants. Some were met at the HC when they came for treatment, while others were interviewed in their homes. In all cases, these interviews were conducted in an

environment that ensured the confidentiality of the respondents' statements, either in a private room at the HC or in a secluded area of the participant's residence. Data collection took place between February and April 2024.

The choice of semi-structured interviews was based on their advantages, notably the ease with which they allow for the collection of useful, in-depth, and relevant information. Because the questions are open-ended, the interviewer can probe further for clarification or obtain more specific answers when necessary [53]. However, the qualitative method can face certain constraints, notably the diversity of information gathered during interviews, which can complicate data analysis. While participant selection followed predefined inclusion criteria to ensure relevance and variation in perspectives, thematic diversity was identified and managed during the coding and analysis process.

Data was collected using an interview guide based on the International Alliance of Patients' Organizations' Declaration of Patient-Centered Care [54]. This guide, pretested in an HA not concerned, and validated by the team members of research, is included in the appendix to this study (S1 Text). It was developed in French and translated into Swahili.

The interviews focused mainly on describing the relationship between patients and providers on the one hand and between patients and the community on the other (particularly support groups). In this study, we focused primarily on the interpersonal and community levels of the social ecological model, reflecting the relationships between patients and providers, and between patients and community-based support structures. The support groups examined in this context were considered to represent the social network level, as they function as peer-based systems of interaction and mutual assistance. Broader societal or policy-level factors were beyond the scope of this investigation. The interviews were especially concerned the providers' respect for patients' needs, autonomy, and independence in the provision of care; the description of patients' involvement in the choice of care strategies about their health and decision-making; their share of responsibility in holistic care; and their ability to support the approach. Whenever possible, the interviews, ended with recommendations for maintaining this BPS care. Participants interviewed at the HCs emphasized partnership and collaboration with providers. Those met at home focused more on family support, emotional experiences, and continuity of care. These differences reflected both institutional and household perspectives of patient involvement.

Two research team members conducted these interviews: the principal investigator (BMK), assisted by another research team member (GJM). These interviews began after obtaining the written and signed consent of the interviewees, while having first explained the purpose of the study. These two researchers are all experienced in conducting qualitative inquiries and have no relationship with the study participants. Before beginning each interview, the researchers introduced themselves to participants and explained that the data were collected exclusively for research purposes. They emphasized voluntary participation and confidentiality to promote openness and trust. The interviewers maintained a neutral and empathetic attitude, listening carefully to participants' experiences without guiding or influencing their responses. To reduce potential bias, interviews were conducted individually in private settings, and participants were reminded not to discuss the interview content with others until the end of data collection. The scheduling and locations of interviews were varied to minimize overlap between participants. As the interviewers were independent of the participants' usual healthcare providers, this approach helped limit both information sharing and social desirability bias, while encouraging honest and spontaneous accounts of personal experiences.

Interviews were conducted in Swahili or in the local language (Shi), according to respondents' convenience. For interviews conducted in the local language, an experienced interpreter, who had no prior relationship with participants and was fluent in both languages, facilitated real-time translation. The same interpreter was used throughout to maintain consistency and minimize variation in interpretation. To reduce potential social desirability bias, the interpreter was trained to remain neutral and avoid influencing participants' responses. Each interview lasted an average of 45 minutes and was audio-recorded. To verify translation fidelity and ensure accuracy of meaning, an independent language expert later reviewed the recordings against the interpreter's translations. All audio files were then transferred to a computer for secure storage.

## Data analysis

During data processing, the translation process was conducted rigorously to ensure both linguistic and conceptual accuracy. The recorded interviews were transcribed verbatim into Word files and then translated into French by two independent translators to ensure fidelity to participants' words. To minimize interpreter bias, the translated transcripts were reviewed and cross-checked by coders for consistency of meaning, and any discrepancies were discussed among translators and coders to preserve the accuracy of participants' intended meanings. Member verification of transcripts was not undertaken, as some authors have questioned its reliability as a validation technique, suggesting that it may alter participants' original accounts or introduce recall bias [55]. Instead, credibility was strengthened through coder and supervisor triangulation, peer debriefing, and back-translation procedures to ensure the accuracy and trustworthiness of the interpreted data.

We analyzed the transcripts using the Inductive Content Analysis approach, as proposed by Vears DF & Gillam L [56]. Given that the BPS model of care is little explored in the DRC, this approach was appropriate for capturing the lived experiences of patients with chronic pathologies, and obtaining practical proposals for care and an application of the study results in the definition of health policies for chronic disease management [57]. The principal investigator (BKM) first familiarized himself with the data by reading and rereading the transcripts. The coding process occurred at three levels: (1) identifying meaning units from participants' narratives and condensed into concise formulations; (2) grouping these condensed units into subcategories based on similarity; and (3) clustering subcategories into broader categories. To ensure credibility and trustworthiness, three other researchers (CENM, SLM, and GJM) joined the principal investigator in reviewing and validating the codes, verifying consistency with the data collected [58]. Their collaborative review led to the identification of additional emergent codes and the refinement of subcategories [59]. Although coding was conducted inductively, the interpretation of categories was subsequently structured according to the three components of our conceptual framework (the individual, the healthcare provider, and the social group) reflecting the Biopsychosocial (BPS) model. This approach allowed data-driven themes to emerge freely while situating them within a relevant conceptual framework that remains underexplored in the DRC context.

The research supervisors then reviewed the analysis carried out, and exchanges with the team were organized to agree on the proposed subcategories and categories, triangulating their points of view with the team's conclusions to ensure credibility. The interpretation of the results was thus carried out in common agreement after some adjustments, considering the study's objective. Our results are presented according to these subcategories and main categories, illustrated by verbatim. The following table (Table 2) illustrates the analysis process.

This research was conducted by the team, from design to presentation of results, including data collection, analysis, and interpretation, under the mentorship of two experienced researchers who supervised it (one from the School of Public Health at the University of Lubumbashi and the other from the Regional School of Public Health at the Catholic University of Bukavu). Discussions helped to harmonize the different points of view and the presuppositions that would influence the results.

## Researcher reflexivity

The research team consisted of public health researchers with expertise in health systems in the Democratic Republic of Congo and qualitative research methods. All members were affiliated with academic institutions in South Kivu and Lubumbashi, but none were involved in the care of the study participants. The team recognized that their professional backgrounds could potentially influence data interpretation; therefore, reflexivity was maintained throughout the study by organizing debriefing meetings after each phase of data collection, comparing interpretations, and triangulating perspectives during the analysis to minimize individual bias.

## Ethics statement and reporting of results

This study was approved by the Ethics Committee of the Catholic University of Bukavu (UCB) under order number UCB/CIES/NC/002/2024, in accordance with the ethical principles of the World Medical Association (WMA) Declaration of

**Table 2. Illustration of data analysis process.**

| Categories | Subcategories | Codes | Definition | Verbatim |
|---|---|---|---|---|
| Patient participation | Involvement in community activities | Patient clubs to support patient care | Peer groups providing psychosocial support, information and mutual help in managing the disease. | *Sometimes, the club President tells us that he went to the HC in the meeting, and the providers asked him how the club's activities were progressing. He told us that we must sensitize others to come to the club.* |
| | | Concerted organization of home visits | Collaborative coordination of home visits between providers, CHWs and patients. | *I am a CHW. We organize home visits with the HN, and often, I go with our club President to see the other members in their households, and there we talk about their health. We are also informed by the nurse who oversees accompanying us, and we participate in HC activities. For example, if there's a visitor, vaccination or construction activities…* |
| | | Sensitize among entourage to ensure follow-up patient's care | Involvement of family and friends in patient care and follow-up | *... that's why we're doing everything we can to reinforce the home visits, to avoid these harmful comments that lead to the patient being discriminated against and feared. People will see that caregivers and other companions or patients approach the patient. With these home visits, we make the rest of the family aware of the patient's follow-up, with respect for appointments or diet, and above all taking the patient directly to the hospital if he or she shows signs of discomfort.* |

Helsinki. All participants voluntarily accepted and signed the written consent form and agreed to take part in the study as well as to the audio recording of their interviews. Participants were free to express themselves in the language of their choice (Swahili or Shi), with the assistance of an interpreter who also signed the confidentiality agreement. To ensure privacy and confidentiality, participants were guaranteed anonymity. The data were stored on a password-protected computer with all identifying information removed. Authorization to conduct the study was also granted by the South Kivu Provincial Health Division. The study and its findings are reported in accordance with the EQUATOR Network's Standards for Reporting Qualitative Research (SRQR) as proposed by O'Brien et al. [60] (S1 Table).

## Results

### General characteristics of participants

A total of 27 patients took part in the study, compared with the planned number, as shown in the table below (Table 3). Of the 27 participants, 11 were interviewers at the HCs.

Table 4 shows the general characteristics of the study participants. Most respondents were female (67%, n = 27); 45% were aged between 45 and 65. Furthermore, 67% had at least attended school (primary or higher), and the majority (41%) were without formal employment (unemployed and housewives). One of the interviewees was a caregiver in one of the HAs covered by the study and a member of the patient club.

### Categories identified in patient involvement in the BPS model of care

From the data analysis, five main categories emerged regarding patient involvement in the biopsychosocial model of care. The identified categories and subcategories relate to the three components of the Conceptual Framework for Patient Involvement in the BPS Model proposed in this study (Fig 2). These categories are: patient participation, patient empowerment and decision-making, support systems, patient education and behavior change, and suggestions for improvement. Table 5 shows these categories, their subcategories, and the corresponding codes.

### Patient participation in healthcare

**Partnership and consideration in caregiving.** Participants reported that they felt valued by healthcare providers and interacted well with them, describing a relationship based on partnership. They noted that providers took time to listen

**Table 3. Persons interviewed based on origin (HA).**

| Health District | Health Area | Planned | Interviewed | Interviewed at Health Center |
|---|---|---|---|---|
| Bagira | Lumu | 5 | 4 | 1 |
| | Nyamuhinga | 5 | 5 | 2 |
| Katana | Birava | 5 | 5 | 1 |
| | Kabushwa | 5 | 5 | 4 |
| Walungu | Bideka | 5 | 5 | 2 |
| | Lurhala | 5 | 3 | 1 |
| Total | | 30 | 27 | 11 |

**Table 4. Respondents' general characteristics.**

| Variable | Number (%) |
|---|---|
| **Gender** | |
| Male | 9 (33) |
| Female | 18 (67) |
| **Age** | |
| <45 | 6 (22) |
| 45 – 65 years | 12 (45) |
| >65 | 9 (33) |
| **Marital status** | |
| Single | 1 (4) |
| Married | 18 (66) |
| Widowed | 8 (30) |
| **Education** | |
| No education | 9 (33) |
| Primary | 5 (19) |
| Secondary | 12 (44) |
| Higher/University | 1 (4) |
| **Occupation** | |
| Unemployed/housewife | 11 (41) |
| Retailer/Seamstress | 4 (15) |
| Farmer/Breeder | 9 (33) |
| Civil servant/teacher | 3 (11) |

to them and allowed them to express their concerns freely, which helped them feel reassured and motivated about their treatments. One participant commented:

*"They take us as their partners and friends because, when we meet in clubs before starting the teachings, we talk with the caregivers/nurses and tell them about our concerns, problems, and everything we would like, and they consider us."* (Patient HA4, 75-years-old)

This sense of partnership was also reflected in the mutual understanding and collaboration between healthcare providers and patients, as illustrated by another respondent: *"... a few days ago, I was talking to him (Head Nurse or HN), and he understood me. I told him: we arrive here at 6 o'clock. You know we don't eat when we have to come here in the morning. Sometimes you let us leave at 12 o'clock, and one of us went into hypoglycemia. I think the HN has already changed that, and now we're treated in the morning so we can get home on time."* (Patient HA1, 56-years-old)

**Table 5. Categories, subcategories, and codes identified in patient involvement in the BPS model of care.**

| Categories (5) | Subcategories | Codes | Relationship between the components of the conceptual framework |
|---|---|---|---|
| 1. Patient participation | Partnership and consideration in caregiving | Good interaction between patients and healthcare providers | AC |
| | | Mutual understanding between healthcare providers and patients | AC |
| | Involvement in community activities | Patient clubs to support patient care | AB |
| | | Concerted organization of home visits | AC |
| | | Sensitize among entourage to ensure follow-up patient's care | AB |
| 2. Patient empowerment and decision-making | Responsibility in care | Self-management of treatment | AC |
| | | Awareness of own health status | AC |
| | Involvement in decision-making and planning | Making choices about treatment options | AC |
| | | Role of support groups in planning | AB |
| 3. Support systems | Access to healthcare services and patient support | Prevention and promotion to support drug therapy | AC |
| | | Link between health status and other determinants | AC |
| | | Entourage and patient clubs as support | AB |
| | Adapting care to individual needs | Patient feedback and satisfaction in care | AC |
| | | The role of psychologists in emotional care | AC |
| 4. Patient education and behavior change | Cognitive skills development | Decide on your treatment | AC |
| | | Identify the origin of a health problem | AC |
| | Social skills development | Interact in support groups | AB |
| | Emotional skills development | How to regulate emotions | AB |
| | | Managing stress | AB |
| | Information sharing | Living with illness | AB |
| | | Importance of advice from healthcare providers | AC |
| 5. Suggestions for improvement of the BPS model | Humanizing care and services | Acting on professional conscience | AC |
| | | Compassion in BPS care | AC |
| | | Availability of drugs and other inputs, group support (sport club) | AC |
| | Psychological and financial support | Availability of psychologists | AC |
| | | Setting up an IGA for financial autonomy | AB |

A = Individual component, B = Social group component, C = Healthcare provider component (Fig 2); IGA = Income Generating Activities.

However, some participants felt that certain healthcare providers lacked consideration and showed negligence towards them, which caused frustration.

*"The only thing that disturbs us is the HN's lack of participation in our activities like other caregivers do. We're even beginning to think that he's discrediting us" (Patient HA3, 67-years-old).*

**Involvement in community activities.** Existing patient support structures or groups (such as diabetic clubs) provided an environment where patients could discuss their health and well-being in relation to their illnesses. These groups played a crucial role in biopsychosocial care, becoming directly involved in monitoring members' care, conducting awareness-raising activities, and providing mutual support.

*"Sometimes, the club President tells us that he went to the HC in a meeting, and the providers asked him how the club's activities were progressing. He told us that we must sensitize others to join the club."* (Patient HA1, 66-years-old woman)

In collaboration with the healthcare providers, the CHWs and club Presidents carried out home visits to patients' households to ensure follow-up care and implement preventive and promotional activities as recommended by providers. Some club members were also CHWs, which further motivated their involvement and strengthened their interaction with healthcare providers.

*"I am a CHW. We organize home visits with the HN, and often, I go with our club President to visit other members in their households, and there we talk about their health. We are also informed by the nurse who supervises us, and we participate in HC activities. For example, if there's a visitor, vaccination or construction activities."* (Patient HA2, 43-years-old woman)

These home visits also served as an opportunity to raise awareness among patient's entourage about the importance of follow-up care and treatment compliance: *"... that's why we do everything we can to strengthen home visits, to avoid harmful comments that lead to patient being discriminated against and feared. People will see that caregivers and other companions or patients approach the patient. Through these home visits, we make the rest of the family aware of the need to follow-up with the patient, respect appointments and diet, and, above all, take the patient directly to the hospital if they show signs of discomfort."* (Patient HA3, 65-years-old woman)

However, some respondents reported a lack of involvement in HC activities, except during awareness-raising sessions led by the psychosocial agent (PSA) at their club meetings.

*"We're with PSA every time. She teaches us a lot and takes part in our activities. We meet with her on the 24th of every month. Apart from that, they [the providers] do many things without involving us..."* (Patient HA3, 54-years-old)

### Patient empowerment and decision-making

**Responsibility in care.**  Patients recognized their responsibility to maintain their health by following caregivers' instructions, particularly regarding the self-management of treatments such as insulin.

*"... it all depends on my behavior. I am called upon to follow instructions. For example, I'm the one responsible for injecting myself with insulin, after receiving the explanations from the caregiver."* (Patient HA1, 58-years-old woman)

Based on the advice and education they received from healthcare providers, patients became aware of and accepted their health conditions, which helped them to achieve psychological stability, as illustrated by this participant:

*"All I have to do is become aware of and accept my illness. Tell myself that it's an illness like any other, and follow the instructions and advice of my caregiver as the person responsible for my health. That's what allows me to control myself first and keep my mind calm."* (Patient HA4, 55-years-old)

**Involvement in decision-making and planning.**  Most respondents perceived the relationship between patients and healthcare providers at the health facility level positive. Patients reported having a degree of autonomy in making decisions about their care while considering the available treatment options, as this respondent explained:

*"There's no need to impose anything on me because I know situations differ. Sometimes I have to be hospitalized and take insulin, and other times it's just tablets. The nurse may suggest insulin, but with its effects on us (asthenia and*

*intense hunger), I suggest they give me tablets instead. They accept and tell me that the best option was insulin...".* (Patient HA1, 68-years-old woman)

Patients were actively involved in choosing treatments in collaboration with their providers. They discussed treatment options, and providers helped them understand the most appropriate choices based on test results and individual needs, as described by one respondent who was a caregiver and a patient club member:

*"... I'm a caregiver and a patient at the same time, a member of the club. I'm in charge (supervising and educating club members) while simultaneously being one of the patients. We discuss how to deal with each case, and I guide them... We start with teachings and then various examinations. I explain things clearly and help them adopt the treatment that's right for them. Depending on the results of the tests, we give them medication."* (Patient and Caregiver HA3, 65-years-old woman)

Another participant highlighted this share decision-making process among the different therapeutic options: *"Your state of health itself will define which treatment to follow (insulin or tablets). But they give clearly explain the medications they recommend and tell us why they are the best option until we understand."* (Patient HA2, 46-years-old woman)

The degree of club involvement in decision-making varied across HAs. In some, respondents reported that the leaders of their support groups (the clubs) participated in planning activities at health facilities, where health actions were discussed with different stakeholders. During club meetings, members discussed activities concerning them and took responsibility for ensuring their follow-up and implementation.

*"There's the President and those who come after him, who talk to the HN about our care. And in our meetings, the President shows us, for example, the tokens the HN gave him in the HC meeting, which will be used to send patients for treatment"* (Patient HA1, 68-years-old woman)

### Support systems

**Access to healthcare services and patient support in BPS care.** In addition to medication, patients benefited from other supportive treatments, including psychological counselling, dietary recommendations, and adapted physical activities.

*"... in addition to the medication we receive, they also recommend that we respect the diet, the meals to be taken, a little sport (walking), small jobs that can help us to have a little sweat..."* (Patient HA5, 53-years-old)

During consultations at care facilities, providers and patients were able to establish a link between the current state of health and other factors, such as education, employment, and family issues, that could influence it and should be considered in BPS care. *"There are times when problems arise in the family, and we think. We tell ourselves [that] this child is misbehaving, or that. Then you think about it all night, and your blood pressure goes up in the morning. And when I go to the HC, the HN asks me if things are going well at home, if I don't have any other problems that are disturbing me..."* (Patient HA1, 56-years-old)

The patient's entourage also played a crucial role in supporting care, offering encouragement and advice, thereby reinforcing BPS management: *"My wife accompanies me through my illness and always encourages me to go to the health center, even if it's just a simple case of malaria. She puts too much effort into it... We also talk to my family about my state of health, and even with some of my neighbors; and they also give me advice, if possible"* (Patient HA1, 56-years-old)

Patients also received significant support from the social structures and groups to which they belonged. They reported that these groups provided great added value to this BPS care, especially regarding psychological and socio-financial aspects, such as financial access to medication when needed.

*"... There's a change because with our social fund if someone gets sick without medicine, we touch it and buy else-where. This club also helps us to be near each other. At first, you may think you're the only one [who is] sick, but when you meet the others, it helps you psychologically. I recently lost my father, and the group really helped me. They also advice on diet, behavior change...". (Patient HA3, 54-years-old)*

**Adapting care to individual needs.** Patients reported that the care they received respected their needs and preferences. Some expressed confidence in healthcare providers and satisfaction with the treatment received. *"Our nurses take our opinions into account when treating us. I can say that I feel free and satisfied with the treatment I receive here at the health center." (Patient HA2, 43-years-old woman)*

They were cared for holistically: physically, psychologically, and socially. The social dimension was particularly visible within the community, through the support groups to which patients belonged. In some care facilities, the hospital psychologist provided psychological support when the psychosocial worker was unavailable, which helped patients feel emotionally better.

*"The providers here treat us well; they don't impose anything on us. They have time to listen to us and talk to us well... They sometimes send us psychologists who give us advice... We feel our emotions calm." (Patient HA4, 36-years-old woman)*

## Patient education and behavior change

**Cognitive skills development.** Patients reported important skills from through the education provided in their clubs, including how to select their treatments according to their health status. *"Sometimes he gives me medicines that I don't like because I don't feel good about them, the little drugs. I told him, I didn't want that. If there wasn't the big one there, I told him to stay with it, I went go buy somewhere else..." (Patient HA6, 64-years-old woman)*

They were also able to identify the sources of disruptions to their health status from the teachings they received from healthcare providers and clubs' facilitators, many of which are related to family problems. *"My [blood] pressure went up because of harmful thoughts about family problems. My children were delinquents..." (Patient HA2, 60-years-old woman)*

**Social skills development.** The support groups helped patients develop social skills through regular discussions and exchanges with peers and facilitators. During club meetings, they gained the ability to interact and discuss their health collectively. *"...we talk to each other in clubs before the caregivers come for the teachings, and we ask questions, or they ask questions to see if we understand correctly... Besides, I'm beginning to regret the others who are sick but hiding in their houses. There is a difference between those who stay at home and us who come here [in the club]. Those who are at home cannot benefit from these teachings..." (Patient HA6, 50-years-old woman)*

**Emotional skills development.** Some respondents described experiences that helped them regulate emotions or manage daily stress. *"My children help me avoid stress, and if they're not there, I distract myself with the radio and avoid the noise and anything that might disturb me..." (Patient HA5, 53-year-old)*

Promotional activities carried out in the community on the recommendation of healthcare providers were also reported to help with emotional regulation.

*"We're asked to do the sport that makes us sweat a little, and that will help us control ourselves, let off steam. For example, I have a friend in M... I wake up, drive to M... and walk home again." (Patient HA1, 56-year-old)*

However, participants suggested that more efforts should be made to promote collective physical activities, such as group sports or exercise sessions, which were identified as areas needing improvement.

**Information sharing.** Patients received information from healthcare providers and social groups that they considered essential to their treatment. They discussed how to live with their illness, both within their groups and with caregivers and CHWs.

*"There are a lot of people who may have diabetes or high blood pressure, but they are ashamed to come here. For me, coming here has helped me regulate food, and I am free to talk to people about my illness. And years go by. In any case, the information I receive here allows me to live better in society." (Patient HA6, 66-years-old woman)*

Some even exchanged experiences with other clubs, and the information shared between groups helped them to thrive.

*"Here recently, our club and [another] organized a meeting with an exchange of experiences." (Patient HA4, 36-years-old woman)*

*"...we discuss it in our club, and we exchange experiences. We learn the lesson together and the measures that help us in our lives. This information helps us to live better in society." (Patient HA3, 52-years-old woman)*

### Suggestions for improving the BPS model

**Humanizing care and services.** To maintain and strengthen biopsychosocial care, respondents emphasized the need to act first on the professional conscience by reinforcing healthcare providers' capacity for compassionate and ethical care, and by raising awareness among those who show indifference toward patients.

*"In the first [place], it is necessary to revive the professional conscience, according to the oath taken, among the caregivers so that they avoid repugnant to the patients, that they give joy/satisfaction to the patients." (Patient HA3, 65-years-old woman)*

*"...Train the staff who take care of us, especially those who neglect us..." (Patient HA6, 64-years-old woman)*

Compassion on the part of healthcare providers was also identified as essential component of comprehensive and integrated care: *"...also, that caregivers take the time to listen to patients, put themselves in their skin and feel that suffering and consider their complaints in full..." (Patient HA3, 67-years-old male)*

Other respondents highlighted the importance of ensuring the availability of medicines, sports clubs, and other resources to prevent disruptions and improve biopsychosocial care: *"We can get sick, and we don't have money, but if there are benefactors who help our health center with drugs, tests, things like that to treat us..." (Patient HA1, 66-years-old woman)*

**Psychological and financial support.** To reinforce biopsychosocial care, respondents expressed the need for more teaching and for greater availability of psychologists to address the psychological aspects of care.

*"That the authorities also strengthen the teachings, that they give us psychologists to follow the patients psychologically, not just the medicines." (Patient HA2, 60-years-old woman)*

Finally, nearly all participants suggested enhancing social funds to support income-generating activities (IGAs), which could contribute to greater financial autonomy and improve patients' quality of life: *"Give the IGAs that will improve the socio-economic standard of living of some patients because the treatment also goes through the diet, but some are unable to respect it or even fail to eat and may even lose their lives." (Patient HA3, 52-years-old woman)*

## Discussion

Our study highlights the importance of patient involvement, informed decision-making, and the vital role of support groups in the success of BPS care. Five main categories emerged: patient participation, empowerment and decision-making, support systems, education and behavior change, and suggestions for improvement. These reflect factors influencing patient involvement, such as provider-patient partnership, shared therapeutic decisions, health education, self-management skills, organizational inclusiveness, and community engagement. Participants also suggested strengthening the model through more humanized care, improved resource availability, enhanced provider capacity, psychological support, and greater financial autonomy. Through these interrelated categories and subcategories, our findings provide insight into the full conceptual framework proposed (Fig 2). They illustrated how the individual component (patients as active participants) is involved in care and interacts with other components, including healthcare providers, health services, family and social groups, patient clubs, and community participation structures, to shape holistic, PCC [38,39]. Each category is discussed below, with its practical, theoretical, and policy implications and connections to relevant literature.

Patient participation in their care is a factor in the success of the care provision. Positive interaction between providers and patients creates a sense of consideration and motivation to become more involved in care. Most of our interviewees reported that they are taken by providers as partners, reassuring them of their personal difficulties. This partnership constitutes one of the determinants of the confidence that users express towards providers and satisfaction [61]. Encouraging such collaboration should therefore be a priority for health systems aiming to implement the BPS model of care. Participants consistently described being listened to, respected, and considered as partners in their care. This perception of partnership enhanced their motivation to adhere to treatment recommendations and improved their overall experience of care. Within the BPS framework, partnership-based relationships are fundamental because they recognize the patient as an active agent in managing their condition rather than a passive recipient of medical interventions. Establishing a climate of mutual respect and trust helps to bridge the gap between biomedical routines and patients' lived realities. Strengthening partnership-based communication between patients and providers enhances shared decision-making, satisfaction, and adherence (core elements of the BPS model). In resource-limited settings, such collaboration fosters accountability, bridges systemic gaps, and advances humanized, person-centered care through shared responsibility for health outcomes. Evidence shows that partnership-based communication builds trust, satisfaction, and adherence [2,61,62], while reviews on person-centered and BPS care [9,39] confirm that shared decision-making and mutual respect are key to effective, humanized care (especially in resource-limited settings).

Empowerment is a core component of the BPS model, positioning patients as active participants rather than passive recipients of care. It involves awareness of one's health, self-management, and adherence to medical advice (supported by knowledge, confidence, and open communication with healthcare providers). Our findings show that empowerment develops through respectful dialogue, shared decisions, and counseling adapted to patients' needs, which foster trust and understanding. Improving care quality requires involving patients in defining strategies and decisions, ensuring empathy and dignity [12,62]. Prior studies similarly highlight that drawing on patients' lived experiences enhances care quality and system responsiveness [5,19,20]. Beyond the individual level, empowerment extends across the social ecological model, shaping self-efficacy, interpersonal relationships, and community engagement. Through autonomy, collaboration, and mutual respect, empowerment contributes to resilient, people-centered health systems. Empowering patients to participate actively in care and decision-making fosters ownership, adherence, and health literacy while reducing dependence on providers. It builds trust and accountability within the health system and should be embedded as a core principle in chronic disease policies to promote resilience and equity in resource-limited settings. This aligns with studies showing that empowerment enhances self-management, adherence, and quality of care through shared decision-making and mutual respect [3,29,40]. Reviews on the BPS and person-centered models [9,39] emphasize empowerment as central to patient agency and system responsiveness, while evidence from chronic disease care [21,22] confirms that fostering autonomy and collaboration strengthens trust, equity, and resilience in health systems.

Support groups (patient clubs) play a crucial role in biopsychosocial care. Working alongside healthcare providers, they support treatment adherence, conduct home visits, and promote patient sensitization within communities. As highlighted by Draper CA et al. [63], patient clubs serve as safe spaces where individuals seek advice, share experiences, and engage in group activities that foster emotional and social support. Integrating such groups (including those promoting physical activity) into national strategies would help achieve Sustainable Development Goals (SDGs) related to chronic disease reduction and Universal Health Coverage (UHC) in fragile contexts like the DRC. Strengthening collaboration between providers and patient clubs is crucial for sustainable, patient-centered care. Health services can engage stable patients as peer educators, leveraging their lived experience to support others. Similar models in other countries have improved care quality and outcomes [64,65]. These experiences should be capitalized, contextualized, and scaled up at the sub- and supranational levels to promote quality care. These clubs easily bring about healthy behavior change [48,66]. In the DRC, CHWs and Club Presidents also play a key role in home visits and community follow-up, aligning with the country's community-based health approach [36,67–69]. Enhancing collective health-promotion activities, such as supervised group exercises, could further improve adherence and well-being, as demonstrated by successful physiotherapy initiatives in Urban Medicalized Health Centers (CSMU) in eastern of DRC [70]. Finally, providers' attention to social determinants (education, employment, and family context) reflects the holistic foundation of BPS care [69,71,72]. Expanding this analysis to inform broader prevention and policy strategies will strengthen chronic disease management and optimize the operationalization of the BPS model across all levels of the health system. Integrating community-based support systems, such as patient clubs and CHW networks, within the formal health system strengthens the biopsychosocial model by connecting psychosocial, behavioral, and clinical dimensions of care. From a social ecological perspective, these networks operate across individual to policy levels, fostering empowerment, adherence, and resilience while promoting sustainable, people-centered chronic disease care in fragile settings. This aligns with evidence showing that community-based support groups enhance adherence, psychosocial well-being, and continuity of care through peer engagement and shared experiences [48,63,66]. Studies on integrated and community health approaches [68,69,73] confirm that collaboration between providers, CHWs, and patient networks operationalizes the BPS model by addressing social determinants and promoting people-centered, sustainable chronic disease care.

In terms of education and behavior changes, our findings show that patient clubs serve as key platforms for health education, where members receive treatment-related information and peer support. Patient education is central to chronic disease management, helping patients understand their condition, adhere to treatment, and strengthen self-care capacities [63]. Support groups further enhance outcomes through shared experiences, mutual assistance, and improved interactions with providers [26,49,66,74]. Healthcare providers actively participate in Club sessions, offering guidance on hygiene, diet, and preventive behaviors. These educational activities foster functional, interactive, and critical components of health literacy [75], while also developing psychosocial skills such as emotional regulation, stress management, and self-awareness. Together, these elements drive behavior change and improve health outcomes [76]. While the benefits of patient education are well recognized [77–79], applying the BPS model requires tailoring educational strategies to each patient's cultural and social context [80]. Contextualized education fosters sustainable behavioral, social, and systemic change in chronic disease care. Continuous health education embedded in patient clubs enhances health literacy and self-efficacy and facilitates sustainable behavior change. Education that combines knowledge, social learning, and emotional support leads to improved outcomes. This is consistent with evidence that patient education and peer learning improve self-management, adherence, and health outcomes in chronic disease care [48,75,77]. Studies on health literacy and psychosocial support [76,78,79] emphasize that combining cognitive, social, and emotional learning fosters lasting behavior change, while BPS-oriented research [80] highlights the need to tailor education to patients' cultural and social contexts for sustainable, person-centered care.

Participants suggested the need for psychological and, more importantly, socioeconomic support to improve care experiences and outcomes. They described organizing themselves into Clubs that manage a social fund to assist

members in need (covering expenses such as medicines and glucose test strips). This initiative helps mitigate financial barriers that often limit access to healthcare, as reported in other studies highlighting difficulties with transportation, treatment adherence, and medical supplies [81]. The need for a more compassionate and patient-centered approach emerged as a key area for strengthening the BPS model. Some participants associated the indifference and lack of empathy shown by certain healthcare providers with psychological distress, dissatisfaction, and reduced trust in care (factors known to undermine therapeutic outcomes). This finding is consistent with evidence that compassion and empathy are central to patients' well-being and perceived quality of care [82,83]. They therefore called for provider sensitization and capacity-building to promote more humanized care. Alongside strengthening the social fund, respondents suggested developing income-generating activities (IGAs) to foster financial autonomy. Evidence supports that humanizing care and promoting patient self-autonomy enhance both access to and quality of healthcare services [84–87]. These elements should be prioritized in the implementation of the BPS model of care. Humanizing care through compassion, listening, and respect for patient dignity (along with economic and psychological support) enhances satisfaction and adherence. Integrating IGAs into patient-club activities can alleviate economic barriers and improve access to medications and services. This aligns with studies showing that empathy, compassion, and respectful communication improve trust, satisfaction, and adherence [82,83,85]. Research on social and economic support [84,86] demonstrates that integrating financial assistance and empowerment initiatives, such as IGAs, enhances access and continuity of care. Together, these findings support that combining humanized, compassionate care with socioeconomic support strengthens the BPS model and improves health outcomes.

The conceptual framework proposed in this work (Fig 2) highlights the complexity, interdependence, and mutual influence of the DRC's health system components, as well as their complementarity in delivering BPS care. Its application can guide policies and strategies for chronic disease prevention and control by clarifying shared responsibilities in providing comprehensive, people-centered care. Moving from a biomedical to a BPS model requires strengthening patient participation, empowerment, community engagement, and compassionate, partnership-based care. Integrating patient clubs, health education, psychosocial support, and economic empowerment into primary care aligns with global recommendations for people-centered services and UHC [6]. Future research should evaluate the scalability and effectiveness of such integrated BPS models in other low-resource settings.

## Limitations and trustworthiness of the study

This study focused exclusively on patients with chronic diseases (diabetes and/or hypertension) enrolled in Patients' Clubs, using convenience sampling. This approach may have introduced selection bias, as club members likely differ from non-members in motivation, engagement, and peer support. Consequently, their experiences may not fully reflect those of patients receiving care outside such programs. Nevertheless, their experiences offered valuable insights into chronic disease patients involvement in the BPS model of care. The sample size limits the generalizability, especially outside the context of BPS model of care. However, the detailed and contextualized description of participants' experiences supports the transferability of the findings to similar primary care settings in low-resource or fragile contexts. The interview setting (HCs or participants' homes) may have influenced the depth and tone of responses. Participants interviewed at HCs might have felt less comfortable expressing critical views toward providers, whereas those interviewed at home could share more freely about personal, family, or psychosocial experiences. Despite this variation, it enriched the diversity and authenticity of the data. Potential effects of education level and response honesty were minimized through careful translation, data triangulation, and researcher reflexivity. Including patients with at least one year of club participation or attendance at a dozen educational sessions strengthened data reliability, while collaborative analysis and experienced supervision enhanced credibility and transferability. This study could serve as a basis for longitudinal or controlled interventions, including larger, more samples to monitor the evolution of empowerment and measure patient involvement in the BPS model.

## Conclusions

This study enabled us to explore the involvement of patients with chronic diseases in choosing their healthcare policies, taking responsibility for holistic care, and supporting the BPS model. The results highlighted the crucial roles of individuals, healthcare providers, their social environment, and particularly patient organizations as support groups in implementing the BPS model. Health systems should take these five categories into account in relation to the three components of the conceptual framework (individuals, providers, and social groups) when defining policies and integrating BPS care, particularly for the prevention and control of chronic diseases in resource-limited countries. Integrating other health determinants into the care process, providing psychological counseling, and enhancing patients' financial autonomy are. These recommendations will improve the quality and accessibility of health services. These findings provide a foundation for developing participatory strategies and longitudinal studies to strengthen person-centered, BPS care in similar contexts.

## Supporting information

**S1 Text. Interview guide.**
(DOCX)

**S1 Table. Standards for Reporting Qualitative Research (SRQR) as proposed by O'Brien et al.**
(DOCX)

## Acknowledgments

The authors thank all inquiry respondents for their participation in the study as well as the Provincial Health Division of South Kivu for facilitating the HD visit for data collection.

## Author contributions

**Conceptualization:** Bertin Mutabesha Kasongo, Christian Eboma Ndjangulu Molima, Samuel Lwamushi Makali, Hermès Karemere, Ghislain Balaluka Bisimwa, Abdon Mukalay wa Mukalay.

**Data curation:** Bertin Mutabesha Kasongo.

**Formal analysis:** Bertin Mutabesha Kasongo, Christian Eboma Ndjangulu Molima, Gérard Jacques Mparanyi, Samuel Lwamushi Makali.

**Investigation:** Bertin Mutabesha Kasongo, Gérard Jacques Mparanyi.

**Methodology:** Bertin Mutabesha Kasongo, Christian Eboma Ndjangulu Molima, Gérard Jacques Mparanyi, Samuel Lwamushi Makali, Pacifique Lyabayungu Mwene-Batu, Albert Mwembo Tambwe, Hermès Karemere, Ghislain Balaluka Bisimwa, Abdon Mukalay wa Mukalay.

**Supervision:** Bertin Mutabesha Kasongo, Ghislain Balaluka Bisimwa, Abdon Mukalay wa Mukalay.

**Validation:** Bertin Mutabesha Kasongo, Christian Eboma Ndjangulu Molima, Samuel Lwamushi Makali, Pacifique Lyabayungu Mwene-Batu, Albert Mwembo Tambwe, Hermès Karemere, Ghislain Balaluka Bisimwa, Abdon Mukalay wa Mukalay.

**Visualization:** Bertin Mutabesha Kasongo, Christian Eboma Ndjangulu Molima, Gérard Jacques Mparanyi, Samuel Lwamushi Makali, Pacifique Lyabayungu Mwene-Batu, Albert Mwembo Tambwe, Hermès Karemere, Ghislain Balaluka Bisimwa, Abdon Mukalay wa Mukalay.

**Writing – original draft:** Bertin Mutabesha Kasongo, Christian Eboma Ndjangulu Molima.

**Writing – review & editing:** Bertin Mutabesha Kasongo, Christian Eboma Ndjangulu Molima, Gérard Jacques Mparanyi, Samuel Lwamushi Makali, Pacifique Lyabayungu Mwene-Batu, Albert Mwembo Tambwe, Hermès Karemere, Ghislain Balaluka Bisimwa, Abdon Mukalay wa Mukalay.

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
