## [Decision Letter · Decision Letter 0]

3 Jun 2025

PGPH-D-25-00933

Patient involvement in the biopsychosocial integrated primary care model: A qualitative study in three health districts of South Kivu, Democratic Republic of Congo.

Dear Dr. Kasongo,

Thank you for submitting your manuscript to PLOS Global Public Health. After careful consideration, we feel that it has merit but does not fully meet PLOS Global Public Health’s publication criteria as it currently stands. Therefore, we invite you to submit a revised version of the manuscript that addresses the points raised during the review process.

The reviewers suggest that the you could focus on conceptualizing the patient engagement. They also suggested using George Engel's 1977 writing as the basis for BPS and not other only authors that have used it. They have identified other elements that I find useful to improving the your submission.

We look forward to receiving your revised manuscript.

Kind regards,

Ferdinand C Mukumbang, PhD

Academic Editor

Journal Requirements:

1. In the online submission form, you indicated that All data used and analyzed in this study are available from the corresponding author on reasonable request. 

3. Uploaded as supplementary information.

2. Please provide separate figure files in .tif or .eps format.

Additional Editor Comments (if provided):

Reviewers' comments:

Reviewer's Responses to Questions

**Comments to the Author**

1. Does this manuscript meet PLOS Global Public Health’s publication criteria?

Reviewer #1: Yes

Reviewer #2: Yes

2. Has the statistical analysis been performed appropriately and rigorously?

Reviewer #1: N/A

Reviewer #2: N/A

3. Have the authors made all data underlying the findings in their manuscript fully available (please refer to the Data Availability Statement at the start of the manuscript PDF file)?

Reviewer #1: No

Reviewer #2: No

4. Is the manuscript presented in an intelligible fashion and written in standard English?

Reviewer #1: Yes

Reviewer #2: Yes

Reviewer #1: I have uploaded an attachment. The authors go through the attached file and make the necessary corrections. In addition to this, the authors have done an impressive job in their research. As a concept, the biopsychosocial integrated primary care model can support vulnerable communities and enhance health and well-being for people.

Reviewer #2: The major weakness is the conceptual framing - and suggestion care made about how this can be strengthened.

Did the study have ethical approval - please add if so or address the issues of confidentiality, data storage etc.

**Do you want your identity to be public for this peer review?** For information about this choice, including consent withdrawal, please see our Privacy Policy

Reviewer #1: No

Reviewer #2: **Yes: ** Olive Cocoman

---

## [Decision Letter · Decision Letter 1]

11 Jul 2025

PGPH-D-25-00933R1

Patient involvement in the biopsychosocial integrated primary care model: A qualitative study in three health districts of South Kivu, Democratic Republic of Congo.

Dear Dr. Kasongo,

Thank you for submitting your manuscript to PLOS Global Public Health. After careful consideration, we feel that it has merit but does not fully meet PLOS Global Public Health’s publication criteria as it currently stands. Therefore, we invite you to submit a revised version of the manuscript that addresses the points raised during the review process.

Please, address the minor comments that the reviewer has provided.

We look forward to receiving your revised manuscript.

Kind regards,

Ferdinand C Mukumbang, PhD

Academic Editor

Journal Requirements:

Additional Editor Comments (if provided):

Reviewers' comments:

Reviewer's Responses to Questions

**Comments to the Author**

Reviewer #1: All comments have been addressed

publication criteria?

Reviewer #1: Yes

3. Has the statistical analysis been performed appropriately and rigorously?

Reviewer #1: N/A

4. Have the authors made all data underlying the findings in their manuscript fully available (please refer to the Data Availability Statement at the start of the manuscript PDF file)?

Reviewer #1: No

5. Is the manuscript presented in an intelligible fashion and written in standard English?

Reviewer #1: Yes

Reviewer #1: The manuscript should be accepted with minor revisions.

**Do you want your identity to be public for this peer review?** For information about this choice, including consent withdrawal, please see our Privacy Policy

Reviewer #1: No

---

## [Decision Letter · Decision Letter 2]

10 Sep 2025

PGPH-D-25-00933R2

Patient involvement in the biopsychosocial integrated primary care model: A qualitative study in three health districts of South Kivu, Democratic Republic of Congo.

Dear Dr. Kasongo,

Thank you for submitting your manuscript to PLOS Global Public Health. After careful consideration, we feel that it has merit but does not fully meet PLOS Global Public Health’s publication criteria as it currently stands. Therefore, we invite you to submit a revised version of the manuscript that addresses the points raised during the review process.

I invite the authors to address further comments that the reviewers have provided.

We look forward to receiving your revised manuscript.

Kind regards,

Ferdinand C Mukumbang, PhD

Academic Editor

Journal Requirements:

1. Please amend your online Financial Disclosure statement. If you did not receive any funding for this study, please simply state: “The authors received no specific funding for this work.”

2. Please update your online Competing Interests statement. If you have no competing interests to declare, please state: “The authors have declared that no competing interests exist.”

Additional Editor Comments (if provided):

Reviewer #1:

Reviewer #3:

Reviewer #4:

Reviewers' comments:

Reviewer's Responses to Questions

**Comments to the Author**

Reviewer #1: All comments have been addressed

Reviewer #3: (No Response)

Reviewer #4: (No Response)

publication criteria?

Reviewer #1: Yes

Reviewer #3: Yes

Reviewer #4: Yes

3. Has the statistical analysis been performed appropriately and rigorously?

Reviewer #1: N/A

Reviewer #3: Yes

Reviewer #4: Yes

4. Have the authors made all data underlying the findings in their manuscript fully available (please refer to the Data Availability Statement at the start of the manuscript PDF file)?

Reviewer #1: No

Reviewer #3: No

Reviewer #4: (No Response)

5. Is the manuscript presented in an intelligible fashion and written in standard English?

Reviewer #1: Yes

Reviewer #3: Yes

Reviewer #4: Yes

Reviewer #1: Reviewer Comments:

The authors have been able to revise the manuscript for acceptability. However, a few revisions are necessary at this point.

Line 643: Please remove the extra two full stops after disease.

Line 658: "That's" is unacceptable in the manuscript. Please correct it to "That is."

References: I noticed some inconsistencies in the reference formatting, particularly with capitalization and style of article titles. Please ensure the entire reference list is revised to align with the journal’s guidelines. (let the references to be uniform).

This is a uniform referencing:

20. Rinaudo CM, Van de Velde M, Steyaert A, Mouraux A. Navigating the biopsychosocial landscape: A systematic review on the association between social support and chronic pain. PLoS One [Internet]. 2025;20(4):e0321750. Available from: https://doi.org/10.1371/journal.pone.0321750

21. Ankomah SE, Fusheini A, Ballard C, Kumah E, Gurung G, Derrett S. Patient-public engagement for health system improvement in sub-Saharan Africa: A systematic scoping review. BMC Health Serv Res. 2021;21(1047):1–16.

Reviewer #3: I am not sure if I reviewed this article before, however, its scope and quality fit PLOS Global Public Health criteria:topic, methodology, and ethical conduct fit the journal’s scope and standards.

Please find below few areas that still need some actions:

1. Data Availability: While your current data availability statement acknowledges ethical restrictions and provides a contact pathway, it falls short of PLOS’s strong expectation for open data. PLOS requires that all data underlying the findings be made fully available without restriction, with rare exceptions for ethical or legal reasons. To strengthen compliance and increase transparency, I recommend:

a) Preparing a minimal, de-identified dataset that does not compromise participant confidentiality. For qualitative research, this could include:

i) The final codebook used in analysis.

ii) A matrix of themes and subthemes with anonymized participant identifiers.

iii) Illustrative, de-identified excerpts supporting each theme.

b) Depositing this dataset in a recognized open repository (e.g., OSF, Zenodo, Figshare) and including the DOI or stable link in your Data Availability Statement.

c) Clarifying restrictions by explicitly stating that full interview transcripts cannot be shared due to confidentiality and ethics approval constraints, but that qualified researchers may request further access through the corresponding author or your ethics committee.

2. Methods clarity & rigor

a) Describe how saturation was assessed (who decided, when, what evidence).

b) Add a brief reflexivity note (research team backgrounds, positionality, steps taken to mitigate bias).

c) Expand on interpreter influence and mitigation (e.g., member-checking, coder checks on translated segments).

3) Results transparency: Insert a few additional verbatim quotes per major theme/subtheme in Table 4 to show analytic grounding.

4) Limitations: Explicitly discuss selection bias/transferability stemming from club-member inclusion and convenience sampling, and any implications for generalisability to non-club patients.

5) Language & copy-editing: Perform a light language polish for minor grammar/phrasing issues to meet “standard English” criterion.

6) Confirm non-duplication: Add a one-sentence statement in the manuscript explicitly confirming that results have not been published elsewhere.

Reviewer #4: Generally

An important topic on the relationship between patients and providers is highlighted by the authors. The paper is conceptualised robustly with an appropriate design. A final grammar check will be needed.

Specifically

Below are the specific feedback;

Abstract

Consideration between providers means what? Clarify this.

mainly concentrate with the humanisation of care.

Five categories not clear, need to enlist them logically.

Any social economic status differences in participants.

Conclude in line with the five categories in the data.

Introduction

Involving patients ----line 57.

Add Social Ecological Model for involvement in care. Line 69 – 108. (individual – social network – community - societal).

Layers of patient behaviour determinants are more than social supports, that is, individual-social networks-community-societal.

More accurately, illustration of study location needed, use of a map is efficient. Line 47.

Add deeper provider attributes, that is, shared compassion, emotional arousal, aesthetic pleasure, simplification, asymmetry. Lines 171-174. Refer to Sadharanikaran model of communication:

Sadharanikaran Model of Communication

https://sadharanikarantheory.blogspot.com/2017/11/sadharanikaran-model-of-communication.html#:~:text=Sadharanikaran%20model%20of%20communication%20(SMC,commonality%2C%20mutual%20understanding%20or%20oneness. Accessed 02 Sept 2025

*Nandita Kapadia-Kundu, “Development and Application of Sadharanikaran Theory to Handwashing Behavior.”

Data collection

Bring out / mention any variations for participants met at HC and those met at home.

How were participants kept from sharing between themselves about study content? Were interviews held simultaneously for each HC? Mention how social desirability bias was controlled?

It is a coding process not a participant selection criteria that determines analysis. Lines 225-226. The second line 226 is not factually correct regarding analysis because as you select participants you don’t know diversity of information.

Lines 231-237 state why only two layers of the social ecological model were investigated in this study? The current investigation was about: patients and providers, patients and community, but did not include patients and social networks, patients and society, state why.

Social desirability bias introduced when an interpreter and an expert were part of the Shi language interview. How was bias managed? But later, you state there was only one interpreter. Which is which? Kindly be consistent.

Talk about the translation process clearly. Line 256.

What were the levels of coding? What were the condensed units? Line 264-274.

Were the final condensed units fitting the BPS model entirely? Were there no emergent condensed units? Can you share them incase they were in the data.

State the criteria for the total participants per health area? Move Table 2 to Results indicating planned and actual participants. Line 301-307. How many participants were interviewed at HC and how many were interviewed at homes?

Discussion

Implication needs to be stated per finding discussed. Then relate implication to relevant literature. Some of the literature is currently available.

Limitations

State differences between participants interviewed at HC and those interviewed at homes.

**Do you want your identity to be public for this peer review?** For information about this choice, including consent withdrawal, please see our Privacy Policy

Reviewer #1: No

Reviewer #3: No

Reviewer #4: **Yes: ** Leonard Bufumbo

---

## [Editor Report · Decision Letter 3]

13 Nov 2025

Patient involvement in the biopsychosocial integrated primary care model: A qualitative study in three health districts of South Kivu, Democratic Republic of Congo.

PGPH-D-25-00933R3

Dear Kasongo,

We are pleased to inform you that your manuscript 'Patient involvement in the biopsychosocial integrated primary care model: A qualitative study in three health districts of South Kivu, Democratic Republic of Congo.' has been provisionally accepted for publication in PLOS Global Public Health.

Best regards,

Ferdinand C Mukumbang, PhD

Academic Editor